

**Exploring the sources of light-absorbing carbonaceous aerosols by integrating observational**
**and modeling results: insights from Northeast China**
Yuan Cheng[1], Xu-bing Cao[1], Sheng-qiang Zhu[2], Zhi-qing Zhang[1], Jiu-meng Liu[1,*], Hong-liang
Zhang[2], Qiang Zhang[3] and Ke-bin He[4]
[1] State Key Laboratory of Urban Water Resource and Environment, School of Environment, Harbin
Institute of Technology, Harbin, 150090, China
[2] Department of Environmental Science and Engineering, Fudan University, Shanghai 200438,
China
[3] Ministry of Education Key Laboratory for Earth System Modeling, Department of Earth System
Science, Tsinghua University, Beijing, 100084, China
[4] State Key Joint Laboratory of Environment Simulation and Pollution Control, School of
Environment, Tsinghua University, Beijing, 100084, China
* Corresponding author. Jiu-meng Liu (jiumengliu@hit.edu.cn).
**Abstract**
Light-absorbing carbonaceous aerosols are important contributors to both air pollution and radiative
forcing. However, their abundances and sources are still subject to non-negligible uncertainties,
which are highly responsible for the frequently-identified discrepancies between the observed and
modeled results. In this study, we focused on elemental carbon (EC) and light-absorbing organic
carbon (i.e., BrC) in Northeast China, a new targeted region of the latest clean air actions in China.
Three campaigns were conducted during 2018–2021 in Harbin, covering distinct meteorological
conditions and emission features. Various analytical methods were first evaluated, and the mass
concentrations of both BrC and EC were validated. The validated BrC and EC measurement results
were then used for source apportionment, together with other species including tracers (e.g.,
levoglucosan). The observation-based results suggested that despite the frigid winter in Harbin, the
formation of secondary organic aerosol (SOA) was enhanced at high levels of relative humidity
(RH). This enhancement could also be captured by an air quality model incorporating heterogeneous



chemistry. However, the model failed to reproduce the observed abundances of SOA, with
significant underestimations regardless of RH levels. In addition, agricultural fires effectively
increased the observation-based primary organic carbon (POC) concentrations and POC to EC ratios.
Such roles of agricultural fires were not captured by the model, pointing to substantial
underestimation of open burning emissions by the inventory. This problem merits particular
attention for Northeast China, given its massive agricultural sector.



## 1. Introduction

Black carbon (BC) and light-absorbing organic carbon, i.e., brown carbon (BrC), are important
contributors to not only haze pollution but also positive radiative forcing (Bond et al., 2013; Laskin
et al., 2015). While their environmental effects are usually predicted by chemical transport and
radiative transfer models, field observational results are necessary to constrain their simulated
spatial distributions and temporal variations (Koch et al., 2009; Samset et al., 2014; Stohl et al.,
2015; Wang et al., 2018; Gao et al., 2022). For example, several studies suggested that to improve
the agreement between simulated and observed BC concentrations, the BC lifetime should be on
the lower end of those assumed in current models (e.g., Samset et al., 2014). However, the
observational data on both BC and BrC are still subject to considerable uncertainties, largely due to
the lack of reference material and method for both species (Baumgardner, et al., 2012; Petzold et al.,
2013; Lack et al., 2014).
The measurement techniques for BC mass typically fall into four categories, i.e., thermal-
optical (Chow et al., 2007; Cavalli et al., 2010), light absorption (Petzold et al., 2005), laser-induced
incandescence (LII; Schwartz et al., 2006) and aerosol mass spectrometric methods (Onasch et al.,
2012). These approaches are based on different measurement principles, depending on the targeted
properties of BC (Petzold et al., 2013). For example, in the thermal-optical method, a particle-laden
filter is heated in an inert (i.e., He) and oxidizing (i.e., $He/O_2$) atmosphere sequentially to volatilize
and combust the deposited carbonaceous components. BC typically evolves after organic matters
due to its higher thermal stability. In addition, BC is strongly light-absorbing and thus its evolution
could lead to a rapid increase of the filter transmittance signal, which is typically monitored in the
spectral range of red light. Then based on the evolution patterns of the carbon and transmittance



signals, BC mass could be determined as the amount of carbon evolving during a specific segment
of thermal-optical analysis (Cavalli et al., 2010). In addition to the thermal-optical method, BC mass
could also be determined based on the aerosol light absorption coefficient (in $Mm^{-1}$; Moosmüller et
al., 2009), carbon ion signals in a mass spectrum measured by a Soot Particle Aerosol Mass
Spectrometer (SP-AMS; Onasch et al., 2012), or the incandescent radiation emitted during fast
heating, boiling and evaporation of BC in a LII instrument (Moteki and Kondo, 2010). The
multitude of measurement principles result in considerable discrepancies in BC results among
different methods, and interestingly, the discrepancies were usually not constant even for the same
study (Buffaloe et al., 2014; Sharma et al., 2017; Corbin et al., 2019; Li et al., 2019; Pileci et al.,
2021; Tinorua et al., 2024). For example, results from the LII and thermal-optical methods were
found to show BC ratios varying between 0.5 and 1.2 for several background sites in Europe, with
unclear reasons for the variability in discrepancies (Pileci et al., 2021).

Similar to BC, different methods co-exist for the measurement of BrC. For example, BrC's

light absorption coefficient is usually determined based on extract of filter sample (Hecobian et al.,
2010) or total aerosol absorption (Yang et al., 2009). Different relationships have been identified
between the results from these two approaches, e.g., strong correlation and close agreement (Zeng
et al., 2022), moderate to strong correlations with considerable differences in the absolute values
(Kumar et al., 2018; Cheng et al., 2021b), and little correlation (Chen et al., 2022). However, factors
responsible for the inconsistent relationships remain poorly understood. In addition, the
measurement of BrC mass is also challenging. This is particularly the case for studies using organic
solvents (e.g., methanol) to extract or isolate BrC. A major difficulty is that the amount of BrC
dissolved in organic solvents could not be directly measured, whereas the indirect approaches are



still under debate regarding the possible artifacts (Yan et al., 2020).

Nonetheless, the measurement methods of BC and BrC require further refinements to provide

more robust constraints on the modelling results. Such efforts are especially necessary for China,
given its more complex emission sources compared to North America and Europe. Here we focus
on Harbin, a representative megacity in Northeast China. With the improvement of air quality in
other regions such as the North China Plain (Xiao et al., 2021; Wang et al., 2023b), Northeast China
was targeted by the national-level clean air policy for the first time in 2021 (State Council, 2021).
This policy, i.e., the *Circular on Further Promoting the Pollution Prevention and Control Battle*,
proposed an ambitious goal of eliminating heavy or severe air pollution events in Northeast China
and other key regions. In addition, Harbin will be hosting the 9[th] Asia Winter Games in February of
2025, which posed another motivation for cleaning the air in Northeast China. However, the
roadmap for air quality improvement was to some extent masked for Harbin as well as other cities
in Northeast China, given that the sources and formation mechanisms of haze pollution were far
from being well understood with limited studies (e.g., Zhang et al., 2020; Wu et al., 2020; Ning et
al., 2022).

This study aimed at understanding the sources of light absorbing carbon in Harbin, based on a

synthesis of field observation and air quality modeling. We started with the coordinated
determination of BrC and BC masses in filter samples, followed by source apportionment using the
validated observational results. Then we used the observation-based BrC and BC source attributions
to constrain the predictions by an air quality model, with focuses on the model vs. observation
discrepancies and the drivers at play. This study provided implications for further efforts to
understand the haze pollution in Northeast China, with respect to both the measurement and



simulation of carbonaceous aerosols.
**2. Methods**
**2.1 Field observation**

A total of 486 $PM_{2.5}$ samples (24-h integrated) were collected at an urban site in Harbin during

three recent campaigns (Table 1). The sampling was performed on the campus of Harbin Institute
of Technology, using a portable sampler (MiniVol; Airmetrics, OR, USA) operated at a flow rate of
5 L/min with quartz-fiber filters (Pall Corporation, NY, USA). For each sample, a half of the filter
was measured for water-soluble inorganic ions and levoglucosan, using a Dionex ion
chromatography system (ICS-5000[+]; Thermo Fisher Scientific Inc., MA, USA). The other half was
cut into two punches for the determination of organic carbon (OC) and elemental carbon (EC, as a
measure of BC mass), using a thermal/optical carbon analyzer (DRI-2001; Atmoslytic Inc., CA,
USA). The first punch was measured directly, while the second punch was immersed in methanol
(Fisher Scientific Company L.L.C., NJ, USA) for an hour without stirring or sonication, dried in air
for another hour, and then analyzed. All the pairs of untreated and extracted punches were measured
deploying the IMRPOVE-A temperature protocol, with selected pairs also analyzed using NIOSH
(Table 1). In addition, wavelength-resolved light absorption coefficients ($b_{abs}$) of the methanol
extracts were quantified using a spectrophotometer (Ocean Optics Inc., FL, USA) coupled with a
2.5m long liquid waveguide capillary cell (LWCC; World Precision Instrument, FL, USA). Samples
strongly impacted by firework emissions ($N = 2, 3$ and 6 for the three campaigns, respectively)
during the Chinese New Year periods were not further investigated in this study. More details of the
field observations were presented in Cheng et al. (2021a and 2022).
**Table 1.** Summary of $PM_{2.5}$ samples involved in this study. *N* indicates the number of samples from
each campaign. For each sample, both the untreated and extracted punches were used for thermal-
optical analysis. $NP_{IMPROVE-A}$ indicates the number of punch pairs analyzed by the IMPROVE-A
temperature protocol. $NP_{NIOSH}$ was defined similarly. The split of OC and EC was based on the
transmittance charring correction for both protocols.

| Measurement period | Main features[a] | $N$ | $NP_{IMPROVE-A}$ | $NP_{NIOSH}$[b] |
|---|---|---|---|---|
| October 16, 2018–April 14, 2019 | Fires in late winter | 180 | 180 | 180 |
| October 16, 2019–February 4, 2020[c] | Humid winter | 112 | 112 | 73 |
| October 17, 2020–April 30, 2021 | Fires in April | 194 | 194 | 86 |

[a] Main features of the campaigns were presented briefly in Figure S1, and described in detail in
Cheng et al. (2021a and 2022).
[b] The selection of samples analyzed by both protocols will be explained in detail in Section 3.2.
[c] The 2019–2010 campaign covered a relatively short period due to the lockdown policy associated
with the outbreak of COVID-19.
**2.2 Air quality modeling**
A revised Community Multi-scale Air Quality (CMAQ) model was used to simulate OC and
EC in Harbin. Compared to the original version (5.0.1), a major update of the revised model was
the addition of new pathways for secondary organic aerosol (SOA) production, i.e., photochemical
and heterogeneous oxidation of isoprene epoxydiols, methacrylic acid epoxide, glyoxal and
methylglyoxal (Ying et al., 2015). Previous studies suggested that the revised CMAQ could
generally reproduce the observed meteorological conditions and $PM_{2.5}$ concentrations on a national
scale in China (Hu et al., 2016a; Wang et al., 2020). However, the model performance remained
inconclusive for $PM_{2.5}$ compositions in specific regions. In this study, the modeling was performed
over East China with a horizontal resolution of $36 \times 36$ km for the 2020–2021 measurement period.
The simulation results were extracted for the grid cell where the sampling site is located at, and then
compared with the observational results.
**3. Results and discussions**
**3.1 Validation of BrC measurement results**
Extracting filter samples by methanol was a common approach to measure brown carbon.





145 While the light absorption by BrC could be readily determined using the methanol extracts, it

146 remains challenging to quantify the mass concentration of BrC, i.e., methanol-soluble OC (MSOC).

147 Unlike water-soluble OC (WSOC), the measurement of MSOC could not be directly done using a

148 Total Organic Carbon analyzer and instead required indirect methods. For example, a four-step

149 procedure was developed by Chen et al. (2017), including drying the methanol extract in a nitrogen

150 flow, re-dissolving the residues in a small amount of methanol (100 μL), spiking a pre-baked filter

151 punch (prepared for thermal-optical carbon analyzer) with a known volume of the new extract (20

152 μL), and measuring the total carbon (TC) in the spiked filter after drying as MSOC. A simpler

153 approach was to determine MSOC as the difference in OC (or TC) concentrations between the

154 untreated and extracted filter punches. This method was initially developed by Chen and Bond

155 (2010), with a substantial concern being the loss of insoluble carbon (e.g., EC) during extraction.

156 However, this artifact was difficult to evaluate, largely due to the lack of reference method for the

157 measurement of EC mass (Petzold et al., 2013).

158  In addition to EC mass, optical attenuation (ATN) retrieved from the carbon analyzer could be

159 an alternative criterion for estimating the extraction-induced loss of insoluble carbon. ATN was

160 calculated as $\ln(I_{final}/I_{initial})$, where $I_{initial}$ and $I_{final}$ indicates filter transmittance signals ($I$) measured

161 at the beginning and end of thermal-optical analysis, respectively. $I_{initial}$ was lower than $I_{final}$ mainly

162 due to the absorption by light-absorbing aerosols (e.g., EC and BrC) and scattering or more

163 specifically backward scattering (Petzold et al., 2005) by the deposited particles (e.g., inorganic ions

164 and non-absorbing OC). Given that $I$ was monitored at a wavelength of 632 nm, only strongly-

165 absorbing BrC could influence $I_{initial}$ and thus ATN through absorption, while SOA could be

166 considered almost non-absorbing (Lambe et al., 2013; Liu et al., 2015, 2016a). Thus we suggest that



(i) decrease of ATN after extraction, if occurred, could be mainly attributed to three possible factors,
including loss of EC, removal of strongly-absorbing BrC and removal of scattering compounds such
as SOA and nitrate; and (ii) if ATN measured by the untreated and extracted filters (i.e., $ATN_{untreated}$
and $ATN_{extracted}$) were largely unchanged, loss of EC should be negligible. In the following
discussions, ΔATN, which is defined as $ATN_{extracted}–ATN_{untreated}$, will be introduced to quantify the
extraction-induced changes of ATN.

In the 2018–2019 campaign, ΔATN were close to zero for some of the samples, whereas for

the remaining ones, ATN typically decreased to varying degrees after the extraction (Figure 1a).
Here we noticed two distinct samples when exploring the ΔATN results (circled in Figure 1a). One
of them showed the most significant decrease of ATN after extraction (with a ΔATN of –0.32) during
the 2018–2019 measurement period, whereas ΔATN was also considerable for the other sample (–
0.25). The two distinct samples were collected successively during January 12–14, 2019. In this
period, relative humidity (RH) stayed above 85%, and both the sulfur oxidation ratio (SOR) and the
nitrogen oxidation ratio (NOR) exceeded 0.2, with record high concentrations of sulfate (~30 μg/m$^3$)
and nitrate (~40 μg/m$^3$) for the 2018–2019 winter. Given the enhanced production of secondary
inorganic aerosols, removal of nitrate by the extraction was a likely cause for the negative ΔATN of
the two distinct samples. Sulfate was not considered here, as it is insoluble in methanol. As another
component that could result in negative ΔATN, SOA could not be directly measured, whereas the
indirect estimating approaches such as the EC-tracer method typically required EC concentration.
We did not predict SOA at this stage, since the EC measurement uncertainties (e.g., the loss of EC
during extraction) had not been comprehensively evaluated. However, similar to SOA, formation of
sulfate and nitrate was contributed by both heterogeneous and gas-phase reactions (Liu et al., 2021;




Wang et al., 2023), indicating that it should be acceptable to reflect the production of secondary
aerosols (including SOA) based on a synthesis of SOR and NOR. In other words, it was very likely
that the atmospheric conditions with elevated SOR and NOR (e.g., January 12–14, 2019) were also
favorable for SOA formation (this inference would be validated in Section 3.3). For the two distinct
samples, therefore, the removal of scattering components, including not only nitrate but also SOA,
was inferred to be highly responsible for the considerable extraction-induced decreases of ATN.

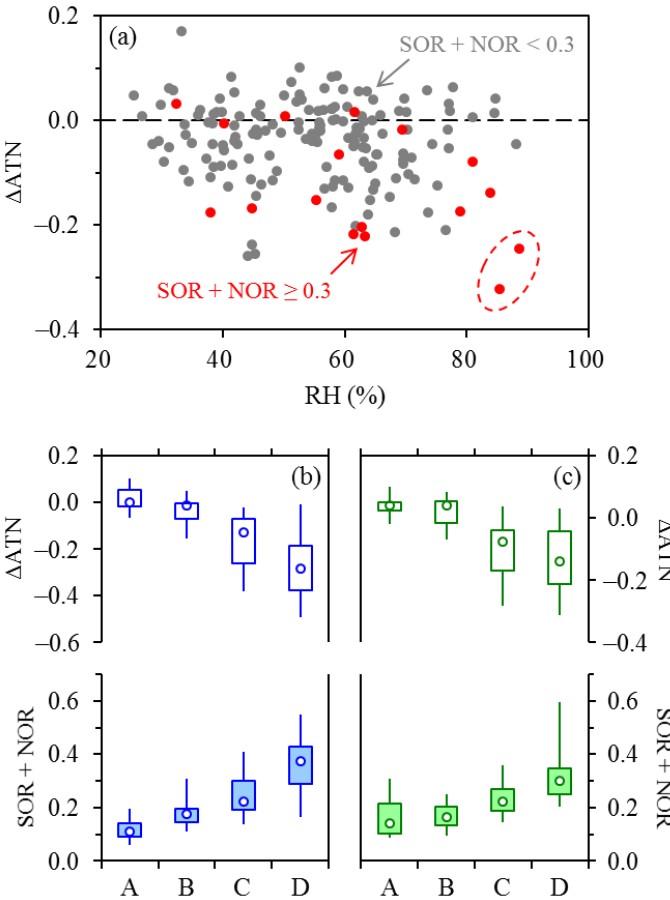


**Figure 1. (a)** Dependence of ΔATN, i.e., $ATN_{extracted}$–$ATN_{untreated}$, on RH during the 2018–2019
campaign, with results in different SOR + NOR ranges distinguished. The dashed line indicates a
ΔATN of zero. The dashed oval highlights two samples characterized by high RH levels, enhanced
formation of secondary aerosols, and considerable decreases of ATN after extraction. **(b)**



Comparisons of ΔATN (upper panel) and SOR + NOR (lower panel) across different RH ranges
(i.e., below 60%, 60–70%, 70–80% and above 80% as indicated by A–D, respectively) during the
2019–2020 campaign. To isolate the role of RH, only the samples with little influence of agricultural
fires were involved in the comparison. In each panel, lower and upper box bounds indicate the 25th
and 75th percentiles, the whiskers below and above the box indicate the 5th and 95th percentiles,
and the open circle within the box marks the median (the same hereinafter). **(c)** The same as (b) but
for 2021–2022.
Regarding the entire 2018–2019 campaign, humid events were actually uncommon, and most
samples with negative ΔATN values concentrated in the conditions with relatively low RH levels of
below 80 % (Figure 1a). Thus, in addition to the enhanced secondary aerosol production at high RH,
there must exist other influencing factors responsible for the change of ATN for the 2018–2019
samples. We then investigated the role of biomass burning, which could emit strongly-absorbing
BrC with mass absorption efficiencies comparable to black carbon (Alexander et al., 2008; Hoffer
et al., 2016; McClure et al., 2020). The 2018–2019 campaign was characterized by frequent
occurrences of agricultural fires (Figure S1), mainly in winter due to a one-off policy which crudely
approved a three-month long period (early December 2018 to early March 2019) for legitimate open
burning. In our previous studies (Cheng et al., 2021a), the fire episodes were identified by the
measured levoglucosan to organic carbon ratios (LG/OC$^*$, where OC$^*$ indicates the untreated OC
based on IMPROVE-A) together with the satellite-based fire hotspots, and the 2018–2019 samples
were classified into three groups with increasing impacts of open burning. In this study, we revisited
the classifications using the levoglucosan to TC ratios (LG/TC), as the TC measurement was
independent of thermal-optical protocol. The classifications made by Cheng et al. (2021a) were
found to still hold, as LG/TC correlated strongly with LG/OC$^*$ ($r = 0.998$; Figure S2). As shown in
Figure 2a, ΔATN were close to zero under little impact of open burning. However, ΔATN turned
negative when the fire impacts were non-negligible, and the negative ΔATN values became more





considerable as the fire impacts increased. For the 2018–2019 campaign, therefore, the occurrences
of negative ΔATN were strongly associated with agricultural fires, e.g., through the removal of BrC
by extraction. In addition, both nitrate and NOR were found to increase with stronger influences of
agricultural fires (Figure S3), presumably due to the enhancement of nitrate production by open
burning emissions (Akagi et al., 2012; Collier et al., 2016; Liu et al., 2016b). Thus, although the
nitrate concentrations (Figure S3) were the lowest for 2018–2019 among the three campaigns, the
removal of nitrate by extraction could also be partially responsible for the association between
ΔATN and agricultural fires.

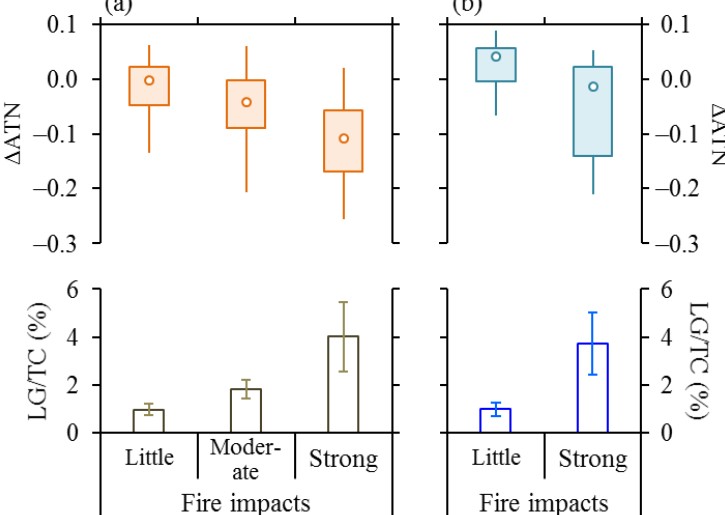


**Figure 2. (a)** Comparisons of ΔATN (upper panel) and LG/TC (on a basis of carbon mass; lower
panel) across three cases with increasing impacts of agricultural fires during the 2018–2019
campaign. To highlight the role of fires, the two distinct samples showing apparent influences of
RH (as circled in Figure 1a) were not involved in the comparisons. **(b)** Comparisons of ΔATN (upper
panel) and LG/TC (lower panel) between two cases with little and strong impacts of agricultural
fires during the 2020–2021 campaign. Only the samples with RH levels of below 70% were
involved, because (i) the influence of RH was insignificant for this RH range and (ii) the majority
of the 2020–2021 samples with strong fire impacts (24 out of 27) fell within this RH range. The
"moderate" case was not identified for 2020–2021. This is mainly because in response to different
policies on open burning, the agricultural fires spanned a relatively long period (more than two
months) during 2018–2019 but concentrated in April during 2020–2021 (Cheng et al., 2022).





Figures 1a and 2a suggest that ATN indeed decreased after the extraction for some of the 2018–
2019 samples. However, the negative ΔATN were found to be associated typically with agricultural
fires and occasionally with high RH conditions. The underlying mechanisms could be attributed
primarily to the removal of BrC and scattering components (including SOA and nitrate), respectively.
Importantly, ΔATN were negligible under little impact of agricultural fires (with a median value of
0.00; Figure 2a), suggesting that the loss of insoluble carbon (e.g., EC) should be negligible during
our extraction procedures.
In addition to the two distinct samples shown in Figure 1a, the connections between ΔATN and
RH could be further confirmed by the 2019–2020 campaign, which experienced much more high-
RH events (mainly in winter) compared to 2018–2019 (Figure S4). As shown in Figure 1b for the
2019–2020 samples with little impact of agricultural fires, the high-RH samples were characterized
by elevated SOR and NOR, pointing to enhanced formation of secondary aerosols (presumably
including SOA). A clear association was also observed between ΔATN and RH. ΔATN were
typically negligible when RH stayed below 70%, showing median ΔATN values of 0.00 and –0.01
for the RH ranges of below 60% and 60–70%, respectively. However, ΔATN deviated more
significantly from zero when RH further increased, e.g., with a median ΔATN value of –0.28 for the
RH range of above 80%. Although some primary organic compounds could also be non-absorbing
at 632 nm, it is unlikely that the abundances or emissions of such species would depend on RH.
Thus, the most possible explanation for the negative ΔATN observed at relatively high RH levels
should be the removal of secondary components (including SOA and nitrate) by extraction.
The 2019–2020 campaign covered a shorter period (Table 1) and encountered much fewer fire
episodes ($N = 2$) compared to 2018–2019 and 2019–2020 ($N = 21$ and 27, respectively). The two



2019–2020 samples with strong fire impacts had similar RH levels of ~50% and only one of them
exhibited considerable ΔATN (–0.26; Figure S5), which could be attributed to the removal of BrC
by extraction. For the 2019–2020 campaign, therefore, the extraction-induced decreases of ATN
were caused primarily by the removal of scattering components. In addition, as shown in Figure 1b,
the close-to-zero ΔATN values observed at the relatively low RH levels (e.g., with a median ΔATN
of 0.00 for the RH range of below 60%) further supported the inference on negligible loss of
insoluble carbon during extraction.

The 2020–2021 campaign experienced more high-RH events compared to 2018–2019 and

more agricultural-fire episodes than 2019–2020 (Figure S1). Correspondingly, the extraction-
induced changes of ATN could be attributed to the removal of either scattering components (Figure
1c) or BrC (Figure 2b). Similar to results from the other two campaigns, ΔATN were close to zero
for the 2020–2021 periods with low RH levels and little impact of open burning (Figure 1c),
demonstrating again that the extraction-induced loss of insoluble carbon was negligible.

The discussions above suggested that it was acceptable to attribute the reduced TC

concentrations in the extracted punches to the dissolving of organic compounds. This supported the
determination of MSOC as the difference in TC between the untreated and extracted punches, i.e.,
$TC_{untreated}$–$TC_{extracted}$. TC was used here since it was independent of analytical method, i.e., not
influenced by the uncertainties in the split of OC and EC. In addition, both $TC_{untreated}$ and $TC_{extracted}$
had been corrected by blanks before being used to calculate MSOC. A total of 53 filters were kept
as blanks for the three campaigns. They exhibited comparable TC loadings before and after the
extraction (averaging 0.61 ±0.23 and 0.44 ±0.21 μgC/cm³, respectively), with no EC detected. This
also indicated that the methanol retained by the filters after the extraction could be completely



volatilized during the drying process, and consequently would not influence the split of OC and EC
for the extracted samples.
**3.2 Evaluation of EC from different methods**
As mentioned in Section 2.1, all the pairs of untreated and extracted punches were measured
by IMPROVE-A, with selected pairs also analyzed by NIOSH. A major purpose of involving
NIOSH was to unfold the EC discrepancies between different protocols, an important indicator for
the EC measurement uncertainties. The 2018–2019 campaign was characterized by intensive
agricultural fires in winter (Figure S1), providing an opportunity to evaluate the effects of open
burning emissions on EC determination. In addition, considering this campaign was the first one to
investigate EC measurement uncertainties in Northeast China, all the 2018–2019 samples were
analyzed by NIOSH (Table 1). The 2019–2020 campaign was characterized by unusually high levels
of RH in winter (Figure S1), which favored heterogeneous chemistry. To investigate the influences
of secondary aerosols on EC determination, NIOSH was applied to all the samples collected in
December 2019 and January 2020. The 2020–2021 campaign showed mixed features of the other
two campaigns, i.e., high RH events and agricultural fire episodes in January and April of 2021,
respectively (Figure S1). Thus all the samples from these two months were analyzed by NIOSH.
For the other periods of 2019–2020 and 2020–2021, NIOSH was used every five samples. As shown
in Table 1, a total of 339 pairs of untreated and extracted punches were analyzed by NIOSH in
addition to IMPROVE-A. Then for the majority of the Harbin samples (339 out of 486), there were
four sets of EC and OC results. Two sets were derived from the untreated punch, using the
IMPROVE-A and NIOSH protocols, respectively. For the third set, EC was measured by the
extracted punch based on IMPROVE-A ($EC_{extracted,\ IMPROVE-A}$) while OC was calculated as the





difference between TC$_{untreated}$ and EC$_{extracted, IMPROVE-A}$. OC and EC of the fourth set were defined
similarly based on NIOSH. The following patterns were observed when comparing the EC and OC
results across different methods.
First, only the OC to EC ratios determined by the untreated samples using IMPROVE-A, i.e.,
(OC/EC)*, could properly reflect secondary aerosol formation. For a typical urban site,
anthropogenic emissions such as those from coal combustion and vehicles were usually considered
relatively stable during a given period, e.g., a specific season. Then the temporal variations of
OC/EC could be used to track SOA formation (e.g., as done by the EC-tracer method), after
excluding the episodes impacted by irregular emissions such as open burning and fireworks. As
firework events were not involved in this study, here we focused on three periods owing all the four
sets of OC and EC results with insignificant influence of agricultural fires, i.e., a four-week long
period in the 2018–2019 winter (December 28, 2018–January 25, 2019), December 2019 to January
2020, and January 2021. In the first case, three samples collected during January 12–15, 2019
exhibited persistently high levels of RH, SOR and NOR (Figure 3), pointing to enhanced formation
of secondary species possibly through heterogeneous chemistry. This humid period is supposed to
favor SOA production as well, since field observational results from the North China Plain
repeatedly showed concurrent increases of secondary inorganic and organic components under high
RH conditions in winter (Hu et al., 2016b; Liu et al., 2020; Sun et al., 2020). Similar to SOR and
NOR, (OC/EC)* also increased substantially for the humid period during January 12–15, 2019
(averaging 12.09 ±0.97) compared to results from the other samples (averaging 5.39 ±1.04; Figure
3). However, unlike (OC/EC)*, OC to EC ratios determined in other approaches (namely OC/EC-I,
-II and -III) less accurately or failed to track the RH-dependent enhancement of SOA formation




(Figure S6). This conclusion also held for the other winters. Briefly, (OC/EC)* increased
concurrently with SOR and NOR at high RH levels for winters of both 2019–2020 (Figure 4) and
2020–2021 (Figure S7), whereas the alternative OC/EC ratios did not.

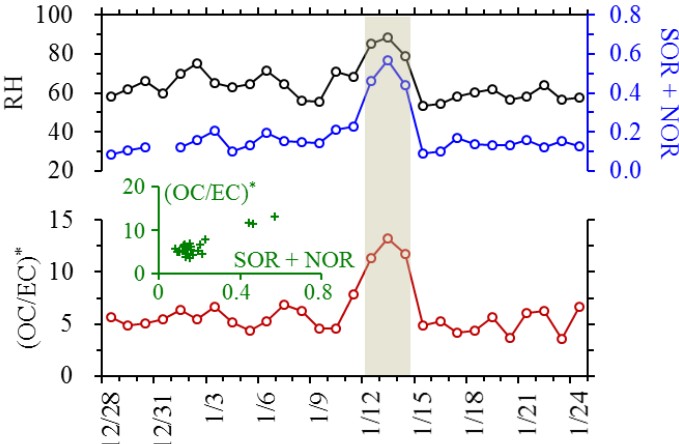


**Figure 3.** Temporal variations of RH, SOR +NOR (upper panel) and (OC/EC)* (lower panel) during
the 2018–2019 winter period with insignificant impact of agricultural fires. The shadowed area
highlights three distinct samples characterized by high RH and enhanced formation of secondary
aerosols. The inner scatter plot shows the positive dependence of (OC/EC)* on SOR + NOR (*r* =
0.89).

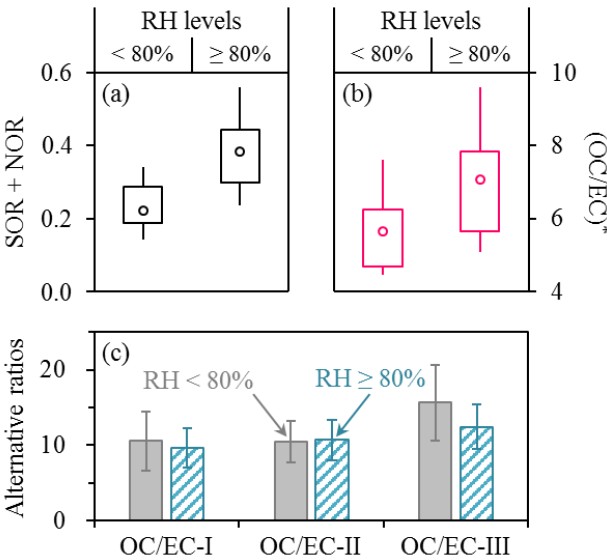


 

**Figure 4.** Comparisons of **(a)** SOR + NOR, **(b)** (OC/EC)* and **(c)** alternative OC/EC ratios between RH ranges of below and above 80%, based on results from December 2019–January 2020. In (c), OC/EC-I was derived from OC and EC measured by the untreated sample using NIOSH. OC/EC-II was calculated by $EC_{extracted, IMPROVE-A}$ and the corresponding OC (i.e., $TC_{untreated}- EC_{extracted, IMPROVE-A}$). OC/EC-III was determined similarly based on NIOSH.

Second, EC measured by the extracted filters ($EC_{extracted}$) were typically lower than results from the untreated ones ($EC_{untreated}$), especially for IMPROVE-A. This pattern should be attributed primarily to EC measurement uncertainties rather than EC loss during the extraction, as the later had been demonstrated to be negligible in Section 3.1. Two influencing factors were identified for the $EC_{extracted}$ to $EC_{untreated}$ ratios based on IMPROVE-A (defined as $R_{IMPV}$). The first one was the relative abundance of sulfate on the extracted filter, which could be estimated by the sulfate to $TC_{extracted}$ ratio (sulfate/$TC_{extracted}$). For the samples with little impact of open burning, $R_{IMPV}$ tended to decrease as sulfate/$TC_{extracted}$ became higher, with generally consistent relationships for the three campaigns (Figure 5a). The median $R_{IMPV}$ was 0.86 when the sulfate/$TC_{extracted}$ ratios were below 1, and decreased to 0.62 for the sulfate/$TC_{extracted}$ range of above 4 (Figure S8). We proposed the following hypotheses for the negative dependence of $R_{IMPV}$ on sulfate/$TC_{extracted}$. We first simplified the remained particles on the extracted filters as a mixture of EC and sulfate, as nitrate and the vast majority of OC were soluble in methanol. Then a key assumption was that sulfate could promote the transmission of laser light through the extracted filters (e.g., by forward scattering; Petzold et al., 2005), indicating that the volatilization of sulfate during the inert mode of thermal-optical analysis could lead to a decrease of the transmittance signal ($I$). Thus in the oxidizing mode, a fraction of EC would be consumed to compensate this decrease (i.e. make $I$ return to its initial value) and consequently, elemental carbon mass would be underestimated by $EC_{extracted}$.

In addition to sulfate/$TC_{extracted}$, another influencing factor for $R_{IMPV}$ was open burning. $R_{IMPV}$



determined for the agricultural-fire episodes were lower compared to results from the periods with
the same sulfate/TC$_{extracted}$ range but little impact of open burning (Figures 5b and 5c). As discussed
in Section 3.1, agricultural fires could be a source for strongly-absorbing BrC. For the untreated
filters, such BrC could be difficult to be properly distinguished from EC by the carbon analyzer used
in this study. Thus, a possible explanation for the reduced $R_{IMPV}$ under strong impacts of agricultural
fires was that open burning emissions could result in overestimation of elemental carbon mass by
EC$_{untreated}$ (i.e., the positive artifact). Under this assumption, the fire-induced decreases of $R_{IMPV}$
could be translated into positive artifacts of ~25% (based on the median $R_{IMPV}$ determined under
little and strong fire impacts) for the open burning episodes of 2018–2019 and 2020–2021.

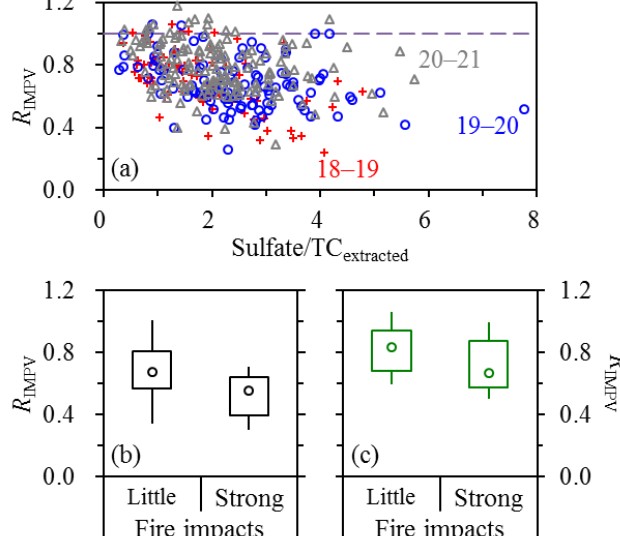


**Figure 5. (a)** Dependence of $R_{IMPV}$ (i.e., the EC$_{extracted}$ to EC$_{untreated}$ ratio based on IMPROVE-A) on
sulfate/TC$_{extracted}$, with results from different campaigns shown separately. Only the samples with
little influence of open burning were involved. **(b)** Comparison of $R_{IMPV}$ between the 2018–2019
samples with strong impacts of agricultural fires (as indicated by "Strong") and those with the same
sulfate/TC$_{extracted}$ range but little fire impact (as indicated by "Little"). **(c)** The same as (b) but for
2020–2021.

Similar to $R_{IMPV}$, the EC$_{extracted}$ to EC$_{untreated}$ ratios based on NIOSH ($R_{NOSH}$) also tended to



decrease with increasing sulfate/$TC_{extracted}$, e.g., with the median $R_{NOSH}$ decreasing from 1.00 to 0.78
as sulfate/$TC_{extracted}$ became higher (Figure S8). Thus, the inference on the underestimation of
elemental carbon mass by $EC_{extracted}$ should be valid for NIOSH as well. The close-to-one $R_{NIOSH}$
but lower $R_{IMPROVE-A}$ (0.86) determined for the same sulfate/$TC_{extracted}$ range of below 1 (Figure S8)
suggested that the extraction led to comparable $EC_{untreated}$ and $EC_{extracted}$ when using NIOSH but
resulted in relatively low $EC_{extracted}$ when using IMPROVE-A. This prohibited the use of $EC_{untreated}$
vs. $EC_{extracted}$ relationship for the assessment of EC loss during extraction, and highlighted the
significance of the ΔATN-based evaluation results in Section 3.1. It is also noteworthy that a
considerable number of samples showed $R_{NIOSH}$ values above 1, indicating that $EC_{untreated}$ was even
lower than $EC_{extracted}$ when analyzing these samples by NIOSH. A possible explanation was that the
NIOSH-based $EC_{untreated}$ also underestimated the elemental carbon mass, i.e., both $EC_{extracted}$ and
$EC_{untreated}$ were biased low (more significantly for the latter) when applying NIOSH to the Harbin
samples. In addition, no evidence was observed for apparent influence of open burning on $R_{NIOSH}$
(Figure S9). It appeared that the determination of $EC_{untreated}$ was less significantly affected by
agricultural fires when using NIOSH compared to IMROVE-A.

The third pattern derived from the comparison of EC results across different methods was that

for the untreated samples, the IMRPROVE-A protocol led to higher EC values than NIOSH (Figure
6). This pattern was in line with results from other regions (e.g., Chow et al., 2004; Piazzalunga et
al., 2011; Giannoni et al., 2016), and was consistent with the previous inference on the uncertainty
of the NIOSH-based $EC_{untreated}$. In addition, the discrepancies between $EC_{untreated}$ measured by the
two protocols became larger with increasing impacts of agricultural fires (Figure 6). This trend could
be attributed to the open-burning-induced overestimation of elemental carbon mass by $EC_{untreated}$



(i.e., the positive artifact), which was considerable for IMPROVE-A (Figures 5b and 5c) but
appeared insignificant when using NIOSH (Figure S9). Another noteworthy feature in Figure 6 was
that compared to the open burning episodes of 2020–2021, the 2018–2019 fire events showed more
significant inter-protocol differences in $EC_{untreated}$. The contrast appeared to indicate that the 2018–
2019 fires, which were inferred to have lower combustion efficiencies (Cheng et al., 2022), could
result in more significant positive artifacts for IMPROVE-A.

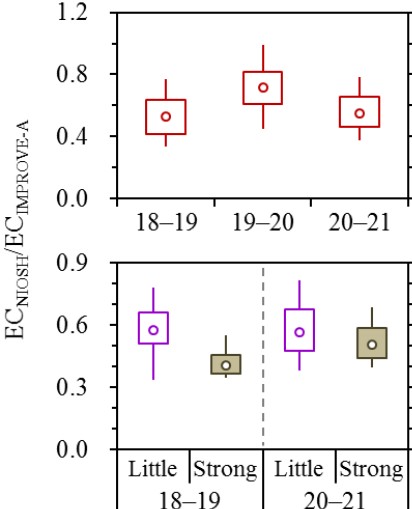


**Figure 6.** Ratios between EC measured by different protocols using the untreated samples, i.e.,
$EC_{NIOSH}/EC_{IMPROVE-A}$. The upper panel compares the ratios across campaigns. The lower panel
compares the ratios between the samples with little and strong impacts of agricultural fires, with
results from 2018–2019 and 2020–2021 shown separately.

As reflected by the discussions above, all the EC results had uncertainties, regardless of the
pretreatment approaches (with or without methanol extraction) and temperature protocols
(IMPROVE-A or NIOSH). For the untreated samples, the IMRPOVE-A protocol led to OC/EC
ratios in reasonable accordance with secondary aerosol formation, whereas NIOSH did not.
However, it must be acknowledged that for IMPROVE-A, the elemental carbon mass was likely





overestimated by $EC_{untreated}$ under strong impacts of agricultural fires (by ~25%), presumably due to
the interference of BrC. Although this positive artifact could in principle be reduced or minimized
by methanol extraction, a new issue arose that the elemental carbon mass was underestimated by
$EC_{extracted}$ (i.e., the negative artifact), which was inferred to be associated with the volatilization of
sulfate from the extracted samples during the inert mode of thermal-optical analysis. The
significance of the negative artifact could be reflected by decreases of EC after extraction, which
were as high as ~15–40% for IMPROVE-A (Figure S8). Importantly, the negative artifact was not
limited to the open-burning-impacted samples, i.e., it also biased the measurement of samples with
little influence of agricultural fires. Thus, although the methanol extraction could reduce the positive
artifacts of $EC_{untreated}$ for the fire episodes, it in turn caused more significant negative artifacts of
$EC_{extracted}$ for all the Harbin samples. Consequently, the methanol extraction was not considered an
effective approach to improve the measurement of elemental carbon mass in this study. In the
following discussions, the $EC_{untreated}$ results based on IMPROVE-A, i.e., $EC^*$, will be used for
exploring the sources of light-absorbing carbon in Harbin.
**3.3 Sources of light-absorbing carbon**

Based on the observational results, EPA's Positive Matrix Factorization (PMF) model (version

5.0) was used to elucidate the sources of light-absorbing carbonaceous aerosols. Here we focus on
the 2020–2021 campaign, which experienced coexisted features of 2018–2019 and 2019–2020 (i.e.,
strong impacts of agricultural fires and high-RH conditions, respectively). A six-factor solution was
resolved by PMF (Figure S10), using time series of $EC^*$, BrC mass concentration (i.e., MSOC),
light absorption coefficient of BrC at 365 nm [i.e., $(b_{abs})_{365}$], levoglucosan, chloride, sulfate, nitrate
and ammonium as inputs. Briefly, two factors were considered secondary due to their dominant



contributions to secondary inorganic ions; two factors were attributed to primary emissions from
biomass burning (BB), as they explained the vast majority of levoglucosan; the last two factors were
important contributors to EC and chloride but had little levoglucosan or secondary species, pointing
to primary emissions from non-BB sources (e.g., coal combustion and vehicles). MSOC apportioned
into these three source categories were termed as sec-MSOC, pri-MSOC$_{BB}$ and pri-MSOC$_{non-BB}$,
respectively. Source-resolved BrC light absorption were defined similarly, as sec-BrC, pri-BrC$_{BB}$
and pri-BrC$_{non-BB}$. EC* emitted by the BB and non-BB sources were referred to as EC$_{BB}$ and EC$_{non-}$
$_{BB}$, respectively. Figure 7 presents an overview of the source apportionment results. The temporal
variations of the MSOC and $(b_{abs})_{365}$ source attributions were characterized by considerable
increases of the BB contribution in April, the season with frequent occurrences of agricultural fires.
It was also noticed that secondary formation was an important source of MSOC (especially in winter)
but contributed less significantly to $(b_{abs})_{365}$. This pattern could be attributed to the fact that
secondary BrC was typically less absorbing than primary BrC (Kumar et al., 2018; Cappa et al.,
2020). For the sources of EC*, a noteworthy feature was that the BB contributions reached similarly
higher levels in the fire-impacted April and January, the coldest month with little influence of open
burning.

The revised CMAQ predicted the concentrations of organic and elemental carbon (i.e., OC$_{mod}$

and EC$_{mod}$), with the primary and secondary OC (i.e., POC$_{mod}$ and SOC$_{mod}$) results also available.
Given that MSOC approximately equaled OC* (Figure S11), it should be acceptable to perform
direct comparisons between these two terms from various sources, i.e., between SOC$_{mod}$ and sec-
MSOC, and between POC$_{mod}$ and primary MSOC (pri-MSOC, calculated as the sum of pri-MSOC$_{BB}$
and pri-MSOC$_{non-BB}$). For the samples with little influence of agricultural fires, the revised CMAQ



generally reproduced the observation-based pri-MSOC and EC$^*$ (Figure 8a), with mean biases of –
1.94 µgC/m$^3$ and −0.43 µgC/m$^3$, respectively. In this case, the POC$_{mod}$ to EC$_{mod}$ ratios also coincided
with the measurement results, i.e., the pri-MSOC to EC$^*$ ratios. For example, the two ratios agreed
with respect to both the absolute values and seasonal variations (Figure 8b). These consistencies to
some extent supported the reliability of the source apportionment results from PMF.

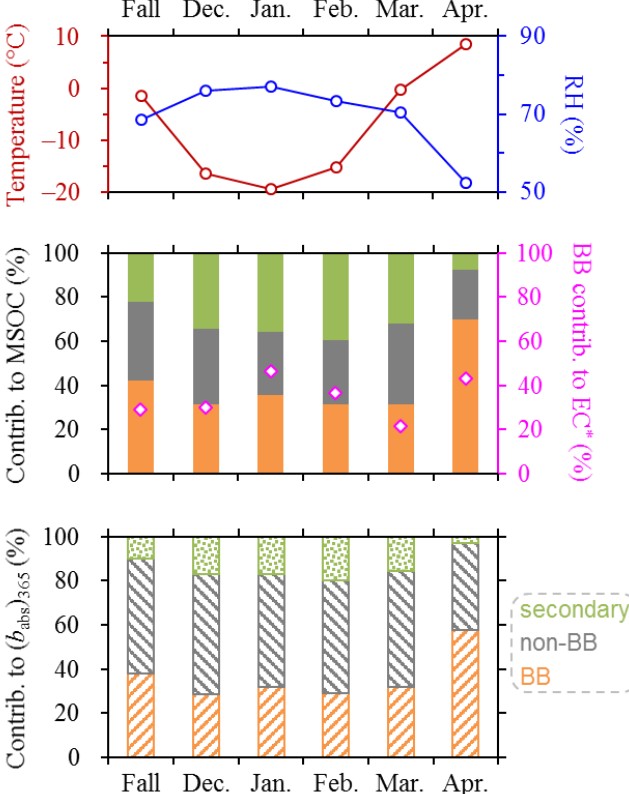


**Figure 7.** Monthly-averaged temperatures and RH (upper panel), and source apportionment results
of MSOC, EC (middle panel) and ($b_{abs}$)$_{365}$ (lower panel) for the 2020–2021 campaign. Fall indicates
mid-October to November. In the middle and lower panels, sources of MSOC and ($b_{abs}$)$_{365}$ were
classified into three categories distinguished by different colors in the bar charts, i.e., primary BB
emissions in orange, primary non-BB emissions in grey and secondary in green. Sources of EC$^*$
were separated into BB and non-BB emissions, with the BB contributions shown by the diamonds
in the middle panel.



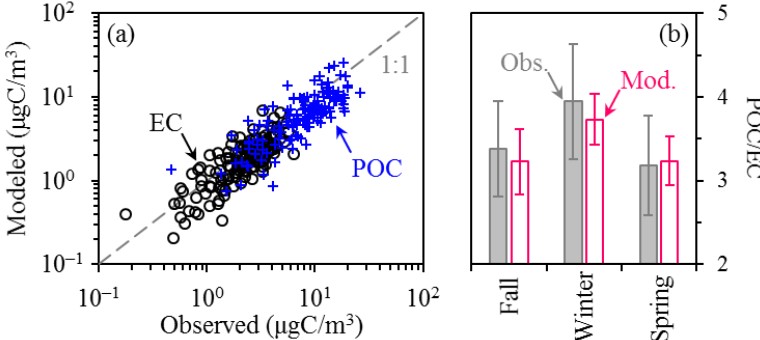


**Figure 8.** Comparisons of the modeled and observed **(a)** POC and EC concentrations, and **(b)** the
seasonal POC to EC ratios for the 2020–2021 campaign. Only the samples with little fire impact
were involved. The 1:1 line is also shown in (a).

The high-RH conditions were concentrated in the winter, i.e., December 2020 to February 2021.
Such conditions were believed to favor SOA production, as indicated by the RH-dependent
increases of SOR and NOR (Figures 1b and 1c). This inference was further confirmed by the PMF
results, as both the sec-MSOC and sec-MSOC/EC$^*$ were considerably enhanced after RH exceeding
80% (Figure 9). The PMF results also confirmed the link between (OC/EC)$^*$ and SOA formation,
given the agreement between sec-MSOC and results from the EC-tracer method ($r = 0.91$; Figure
S12). The revised CMAQ predicted the RH-dependent enhancement of SOC formation as well.
However, it failed to reproduce the observed SOC concentrations and SOC to EC ratios, with
significant underestimations. For example, the modeling results only explained 18% and 22% of the
observed SOC concentrations (corresponding to 19% and 26% of the observed SOC to EC ratios)
for the RH ranges of below and above 80%, respectively. The results suggested that the SOA module
of the revised CMAQ, including the newly-added heterogeneous mechanisms, still required
substantial improvements.




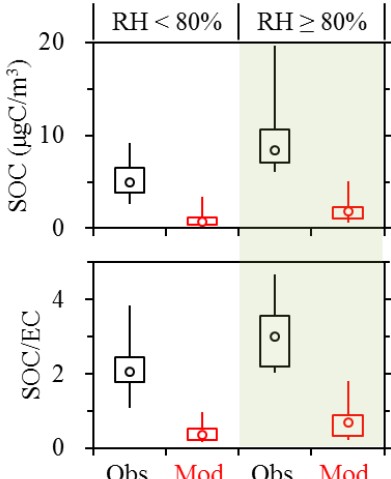


**Figure 9.** Comparisons of the modeled and observed SOC concentrations (upper panel) and SOC
to EC ratios (lower panel) for the 2020–2021 winter. The comparisons were performed for the RH
ranges of below and above 80% separately. Open burning impact was negligible for this period.

The agricultural fire episodes mainly occurred in April during the 2020–2021 measurement
period. PMF results suggested that the BB contributions to MSOC and EC* increased significantly
for the fire episodes (reaching 72 and 44%, respectively) compared to other periods in spring (33
and 25%, respectively). The fire emissions also significantly increased the observation-based POC
concentrations (i.e., pri-MSOC) and POC to EC ratios (i.e., pri-MSOC/EC*; Figure 10). This is
within expectation, since organic compounds were frequently found to constitute the vast majority
of the particulate emissions from open burning emissions (Hodgson et al., 2018; Garofalo et al.,
2019). Since the revised CMAQ did not predict biomass burning OC separately, comparison of the
modeling and observational results could only be made based on the bulk primary OC. As shown in
Figure 10, the model could not track the influences of agricultural fires on primary OC, e.g., as
indicated by the largely comparable POC$_{mod}$ to EC$_{mod}$ ratios between the fire episodes and other
periods in spring. It appeared that the fire emissions, which were derived from the FINN inventory,
were underestimated for the model simulation. In FINN, the open burning emissions were retrieved



using burned areas detected by the Terra and Aqua polar-orbiting satellites (Wiedinmyer et al., 2011).
A limitation of this approach was the missing of fires due to satellite overpass timing (Uranishi et
al., 2019), which was also the case for the Global Fire Emissions Database (GFED), another
commonly-used open burning inventory based on burned areas (Konovalov et al., 2018; Chen et al.,
2023). Previous studies suggested that the underestimation of open burning emissions by FINN or
GFED could be considerable, e.g., by a factor of as high as above 20 (Xie et al., 2024). Given the
massive agricultural sector in Harbin and surrounding areas (e.g., the Harbin-Changchun
metropolitan area), we suggest that the uncertainties of open burning inventories merit particular
attention for the modeling studies in this region.

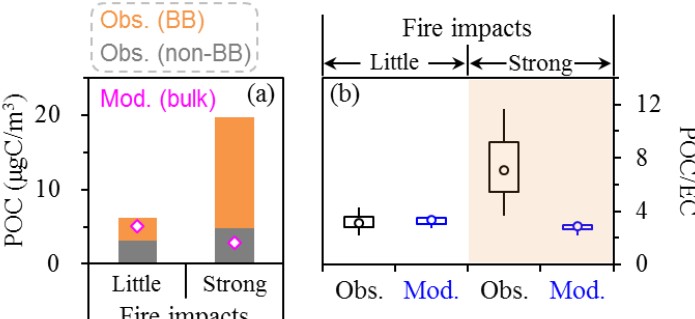


**Figure 10.** Comparisons of the modeled and observed **(a)** POC concentrations and **(b)** POC to EC
ratios between the samples with little and strong fire impacts in the spring of 2021. In (a), the
observation-based results were shown by the bars (as the sum of BB and non-BB emissions), while
the modeling results were indicated by the diamonds.
**4. Conclusions and atmospheric implications**

Light-absorbing carbonaceous aerosols were investigated for Northeast China based on three

campaigns conducted during 2018–2021 in Harbin. BrC masses were determined based on methanol
extraction of filter samples, as the difference between TC concentrations of the untreated and
extracted punches. A long-standing concern on this method was the loss of EC during extraction.



This artifact was evaluated indirectly based on the extraction-induced changes of ATN, due to the
lack of reference method for EC measurement. For different campaigns, it was repeatedly observed
that ATN was largely unchanged after extraction, as long as the RH levels were unfavorable for
secondary aerosol formation and the impacts of agricultural fires were negligible. This pointed to
negligible loss of EC during extraction and consequently supported the robustness of the
observational data on BrC mass. In addition, EC and OC concentrations were determined by four
methods differing with respect to pretreatment approaches (with and without extraction of the filter
samples) and thermal-optical protocols (IMPROVE-A and NIOSH). Results from the untreated
samples using IMPROVE-A were found to provide OC to EC ratios in reasonable accordance with
secondary aerosol formation. Thus EC determined by this method was used for the source
apportionment of light-absorbing carbon, together with other input species such as BrC mass, BrC
absorption coefficient and levoglucosan.

The observation-based source apportionment results showed increased contributions of

secondary formation to BrC in winter, when the high-RH conditions concentrated. It was also
noticed that secondary formation contributed more significantly to BrC mass than BrC absorption,
in line with the consensus that secondary BrC was typically less absorbing than primary BrC. In
addition, agricultural fires were found to effectively enhance the BB contributions to BrC (in terms
of either mass concentration or absorption coefficient) and EC.

The abundances and sources of OC and EC were also predicted by an air quality model with

newly-added heterogeneous reactions. The general equivalence of BrC and OC masses supported
direct comparisons of the observational and modeling results. The model properly reproduced POC
and EC (in terms of both absolute concentration and POC to EC ratio) for the periods with little



impact of agricultural fires. The model also predicted the existence of RH-dependent enhancement
of SOC production in winter, but significantly underestimated the observed SOC concentrations.
Another problem identified for the modeling results was the substantial underprediction of POC for
the agricultural fire events, presumably due to underestimation of open burning emissions by the
FINN inventory.
An agreement between observed and simulated results (e.g., with respect to aerosol abundances
and sources) is essential for the development of efficient air pollution control strategies. In this study,
we constrained the modeling results of carbonaceous aerosols by field observation, based on
validated measurement results of BrC and EC. Two challenges were identified for the simulation of
carbonaceous aerosols in Northeast China, i.e., significant underprediction of SOC and agricultural
fire emissions. Our results suggest that the commonly-used CMAQ model requires substantial
improvements for the application in Northeast China.
**Data availability.** Data are available from the corresponding author upon request
(jiumengliu@hit.edu.cn).
**Author contributions.** YC and JL designed the study and prepared the paper, with inputs from all
the co-authors. XC and ZZ carried out the experiments. SZ and HZ performed the simulations. QZ
and KH validated the results and supervised the study.
**Competing interests.** At least one of the (co-)authors is a member of the editorial board of
Atmospheric Chemistry and Physics.
**Disclaimer.** Publisher's note: Copernicus Publications remains neutral with regard to jurisdictional
claims in published maps and institutional affiliations.
**Acknowledgements.** The authors thank Zhen-yu Du at the National Research Center for



Environmental Analysis and Measurement and Lin-lin Liang at the Chinese Academy of
Meteorological Sciences for their help in sample analysis.
**Financial support.** This research has been supported by the National Natural Science Foundation
of China (grant no. 42222706), and the Fundamental Research Funds for the Central Universities.

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
