# Peer review of "Exploring the sources of light-absorbing carbonaceous aerosols by integrating observational"

_EGUsphere, 2024_

## Author Comment (AC1)

*Comments from Reviewer #1*

*General comments*

This manuscript investigated the sources of black and brown carbon in Northeast China, by integrating observational and simulation results. An agreement between observed and modeled $PM_{2.5}$, especially with respect to the source-resolved chemical compositions, is essential to design efficient air pollution control strategies. Such comparisons have rarely been made for Northeast China, which possessed distinct primary sources and meteorological conditions compared to other regions in China. The authors observed efficient SOA formation at low temperatures during winter and abundant open-burning POA in spring, both of which could not be properly reproduced by CMAQ. The results contribute to the understanding of haze pollution in China. I think this manuscript could be considered for publication after addressing my following concerns.

*Major comment*

My major concern is that for the comparison of observational and modeling results, an interesting point was missing, i.e., the relationship between $EC_{mod}$ and $EC_{obs}$ during the fire episodes. The authors argued that open burning emissions and thus $EC_{mod}$ were underestimated, whereas $EC_{obs}$ were biased high due to fire-induced BrC. Then it would be expected that $EC_{mod}$ and $EC_{obs}$ should have larger differences for the fire episodes compared to other periods, but the authors did not show the comparison. If this inference did not hold, the major conclusions would be questionable.

**Our responses:** We thank the reviewer for the suggestion. In the revised manuscript, $EC_{mod}$ and $EC_{obs}$ were compared for the fire episodes as suggested, and their discrepancies were indeed more significant than the other periods in spring **(see lines 542-547)**:

"*It was also noticed that the mean bias in elemental carbon ($EC_{mod} – EC^*$) was more significant for the fire episodes (–1.26 µgC/m$^3$) compared to other periods in spring (–0.44 µgC/m$^3$). This pattern could be attributed to two factors, including the underestimation of open burning emissions by the inventory and the fire-associated overestimation of elemental carbon mass by $EC^*$. In other words, both $EC_{mod}$ and $EC^*$ were subject to larger uncertainties for the fire episodes, resulting in more significant model vs. observation discrepancies in elemental carbon concentration*".

*Minor comments*

**(1)** Lines 16-18. I guess what the authors want to say is "the understanding on the abundances and sources of light-absorbing carbon is still subject to non-negligible uncertainties". In other words, the "abundances and sources" themselves don't have uncertainties. This sentence needs to be re-organized.

**Our responses:** The sentence was rewritten as suggested: "*However, their abundances and sources remain poorly constrained, as can been seen from the frequently-identified discrepancies between the observed and modeled results*" **(see lines 16-18)**.

**(2)** Line 92. It should be "light-absorbing".

**Our responses:** The change was made as suggested **(see line 95)**.

**(3)** Line 206. I assume the authors mean "2020-2021", not "2021-2022".

**Our responses:** The typo was corrected **(see line 210)**.

**(4)** Line 223. Provide quantitative result for "close to zero".

**Our responses:** The change was made as suggested **(see line 227)**.

**(5)** Line 287. Why did TC decreased after extraction for the blank filters? Is the difference significant?

**Our responses:** The blank TC decreased slightly after the extraction (from $0.61 \pm 0.23$ to $0.44 \pm 0.21$ $\mu gC/cm^3$; the difference was statistically significant), with no EC detected for either the untreated or extracted filters. A possible explanation for the decrease was the dissolving of organic compounds, which constituted the TC of the untreated blank filters, into the solvent. The discussions above were incorporated into the revised manuscript **(see lines 290-295)**.

**(6)** Line 392. Provide quantitative description for "a considerable number".

**Our responses:** The change was made as suggested **(see line 399)**.

**(7)** Line 467. Clarify whether the mean bias was calculated as model – observation.

**Our responses:** The change was made as suggested **(see line 476)**.

**(8)** Figure 9. It would be better to use log scale for the y-axis of the upper panel.

**Our responses:** The change was made as suggested **(see lines 511-514)**:

[Figure]

**Figure R1.** Comparisons of the modeled and observed SOC concentrations (upper panel) and SOC to EC ratios (lower panel) for the 2020–2021 winter. The comparisons were performed for the RH ranges of below and above 80% separately. Open burning impact was negligible for this period. This figure was presented as Figure 9 in the revised manuscript.

**(9)** Caption of Figure S12. $OC^*$ and $EC^*$ should be used for the equation of EC-tracer method. In addition, the firework periods should be clearly shown in the figure, i.e., clarify whether all the events without SOC results were associated with fireworks.

**Our responses:** The figure was revised as suggested:

[Figure]

**Figure R2.** Variations of SOC derived from different approaches for the 2020–2021 winter. SOC was determined as secondary MSOC (i.e., sec-MSOC) based on PMF. In addition, SOC was also estimated by the EC-tracer method as $OC - EC \times (OC/EC)_{min}$, where $(OC/EC)_{min}$ indicates the minimum OC to EC ratio; OC and EC results measured by the untreated samples deploying IMPROVE-A were used for the calculation. SOC resolved by the two approaches showed similar patterns of temporal variation and comparable mass concentrations, leading to a strong linear correlation ($r = 0.91$). As indicated by the shadowed periods, SOC was not estimated for the samples

strongly impacted by firework emissions during the Chinese New Year Period. This figure was presented as Figure S12 in the revised manuscript.

---

## Author Comment (AC2)

*Comments from Reviewer #2*

*General comments*

The manuscript by Cheng et al. explored the characteristics and sources of carbonaceous aerosols during three successive winters in Harbin. Samples were collected and analyzed for a variety of species, e.g., brown carbon, elemental carbon and levoglucosan. The authors evaluated the loss of EC during methanol extraction of filter samples, a long-lasting debate on the measurement method for BrC mass concentration. This artifact was suggested to be unimportant based on indirect evidences, providing valuable implications for future studies. The authors also explained the EC discrepancies between different analytical methods, and identified the OC/EC ratios (i.e., OC and EC results) that were in reasonable accordance with secondary aerosol formation. The authors then performed source apportionment of BrC and EC using the measurement results, and finally compared the observation-based attributions with those predicted by an air quality model. Overall, the results were properly interpreted and presented. However, as listed below, there are some major concerns on top of writing problems.

*Major comments*

**(1)** The authors only broadly stated that the high RH conditions in winter should favor heterogeneous formation of secondary aerosols. This statement could be more specific. For example, did the heterogeneous reactions occur in aerosol water like Beijing or on the surface of frozen particles due to the low temperature?

**Our responses:** We expanded the discussions on heterogeneous chemistry as suggested: "*In addition, aerosol water could remain supercooled at the typical temperatures during winter in Harbin, which were down to about –25 °C in terms of daily average (Rosenfeld and Woodley, 2000). For the frigid atmosphere in Northeast China, therefore, heterogeneous reactions in aerosol water were expected to prevail as long as RH reached sufficiently high levels. The mechanisms of low-temperature chemistry, which may differ from those in the relatively warm regions (e.g., Beijing), merit further investigations*" **(see lines 505-510)**.

**(2)** Line 300. Suggesting toning down the statement, as this study did not have robust evidence for heterogeneous chemistry. Please note that the RH-dependent increases of SOR + NOR (Figures 1b and 1c) and SOC/EC (Figures 3 and 4) should only be

considered indirect evidences.

**Our responses:** We agree with the reviewer that this study lacks direct observational evidence for heterogeneous chemistry. In the revised manuscript, the sentence was rewritten as: "*The 2019–2020 campaign was characterized by unusually high levels of RH in winter, which were expected to favor heterogeneous chemistry*" **(see lines 306-307)**.

**(3)** Please clarify whether it was acceptable to approximate MSOC as untreated OC. This point is important, given that thermal-optical analysis of the extracted filters could be laborious and time-consuming.

**Our responses:** We thank the reviewer for the suggestion. Our results suggested that it was acceptable to approximate MSOC as untreated OC measured by IMPROVE-A. This point was clarified in the revised manuscript: "*Results from the untreated samples using IMPROVE-A were found to provide OC to EC ratios in reasonable accordance with secondary aerosol formation......the corresponding OC (OC$^*$) approximately equaled MSOC, the determination of which was laborious. This equivalence supported the simplification of MSOC as OC$^*$ for further studies*" **(see lines 560-566)**.

*Minor comments*

**(1)** Line 18. Suggest clarifying that EC is a measure of black carbon.

**Our responses:** The change was made as suggested **(see line 19)**.

**(2)** Lines 25 and 27. Change SOA to SOC, as no SOA result was presented throughout the manuscript.

**Our responses:** The change was made as suggested **(see lines 26 and 28)**.

**(3)** Line 77. Suggest providing an example for the "possible artifacts".

**Our responses:** An example was provided as suggested: "*when determining BrC mass as the difference in total carbon concentration between untreated and extracted filters, the result could be biased high due to the loss of insoluble BC during extraction*" **(see lines 78-80)**.

**(4)** Line 102. Full name should be given for PM$_{2.5}$.

**Our responses:** The change was made as suggested **(see line 105)**.

**(5)** Line 142. Use "Results and discussion".

**Our responses:** The change was made as suggested **(see line 145)**.

**(6)** Line 160. Suggest adding a "the" before "filter".

**Our responses:** The change was made as suggested **(see line 163)**.

**(7)** Line 183. Suggest changing the first "as" to "since".

**Our responses:** The change was made as suggested **(see line 186)**.

**(8)** Line 249. Suggest re-writing this sentence as "Importantly, as shown in Figure 2a, ΔATN were negligible……"

**Our responses:** The sentence was rewritten as: "*ΔATN were negligible after excluding these two distinct cases (Figure 2a), suggesting that the loss of insoluble carbon (e.g., EC) should be minimal during our extraction procedures*" **(see lines 253-255)**. We think this statement was more precise than the original description.

**(9)** Line 281. Suggest adding an "in turn" before "supported".

**Our responses:** The change was made as suggested **(see line 285)**.

**(10)** Line 325. Change "is" to "was".

**Our responses:** The change was made as suggested **(see line 332)**.

**(11)** Line 334. Suggest adding a "the" before "winters".

**Our responses:** The change was made as suggested **(see line 341)**.

**(12)** Lines 394-396. The statements did not hold for all the Harbin samples.

**Our responses:** The sentences were rewritten as: "*It is also noteworthy that for NIOSH, ~40% of the samples showed $R_{NIOSH}$ values above 1, indicating that their $EC_{untreated}$ was were even lower than $EC_{extracted}$. A possible explanation was that when using NIOSH, the NIOSH-based $EC_{untreated}$ also frequently underestimated the elemental carbon mass, and the underestimation could be more significant than that by $EC_{extracted}$*" **(see lines 398-404)**.

**(13)** Line 480. Remove the "the" before "seasonal".

**Our responses:** The change was made as suggested **(see line 490)**.

**(14)** Line 542. Suggested adding a "(i.e., MSOC)" after "BrC mass".

**Our responses:** The change was made as suggested **(see lines 563-564)**.